# Integrating Genetic and Chromosome Maps of *Allium cepa*: From Markers Visualization to Genome Assembly Verification

**DOI:** 10.3390/ijms231810486

**Published:** 2022-09-10

**Authors:** Aleksey Ermolaev, Natalia Kudryavtseva, Anton Pivovarov, Ilya Kirov, Gennady Karlov, Ludmila Khrustaleva

**Affiliations:** 1Laboratory of Applied Genomics and Crop Breeding, All-Russian Research Institute of Agricultural Biotechnology, Timiryazevskay 42 Str., 127550 Moscow, Russia; 2Center of Molecular Biotechnology, Russian State Agrarian University-Moscow Timiryazev Agricultural Academy, Timiryazevskay 49 Str., 127550 Moscow, Russia; 3Plant Cell Engineering Laboratory, All-Russian Research Institute of Agricultural Biotechnology, Timiryazevskay 42 Str., 127550 Moscow, Russia; 4Laboratory of Marker-Assisted and Genomic Selection of Plants, All-Russian Research Institute of Agricultural Biotechnology, Timiryazevskay 42 Str., 127550 Moscow, Russia; 5Moscow Institute of Physics and Technology, 141701 Dolgoprudny, Russia; 6Department of Botany, Breeding and Seed Production of Garden Plants, Russian State Agrarian University-Moscow Timiryazev Agricultural Academy, Timiryazevskay 49 Str., 127550 Moscow, Russia

**Keywords:** Tyr-FISH, integration genetic, cytogenetic and pseudochromosome maps, transcript based markers, genome assembly, bioinformatics, *Allium cepa*

## Abstract

The ability to directly look into genome sequences has opened great opportunities in plant breeding. Yet, the assembly of full-length chromosomes remains one of the most difficult problems in modern genomics. Genetic maps are commonly used in de novo genome assembly and are constructed on the basis of a statistical analysis of the number of recombinations. This may affect the accuracy of the ordering and orientation of scaffolds within the chromosome, especially in the region of recombination suppression. Moreover, it is impossible to assign contigs lacking DNA markers. Here, we report the use of Tyr-FISH to determine the position of the short DNA sequence of markers and non-mapped unique copy sequence on the physical chromosomes of a large-genome onion (*Allium cepa* L.). In order to minimize potential background masking of the target signal, we improved our earlier developed pipeline for probe design. A total of 23 markers were located on physical chromosomes 2 and 6. The order of markers was corrected by the integration of genetic, pseudochromosome maps and cytogenetic maps. Additionally, the position of the *mlh1* gene, which was not on the genetic map, was defined on physical chromosome 2. Tyr-FISH mapping showed that the order of 23.1% (chromosome 2) and 27.3% (chromosome 6) of the tested genes differed between physical chromosomes and pseudochromosomes. The results can be used for the improvement of pseudochromosome 2 and 6 assembly. The present study aims to demonstrate the value of the *in situ* visualization of DNA sequences in chromosome-scaffold genome assembly.

## 1. Introduction

The rapid development of DNA sequencing technology has made it possible to produce inexpensive whole-genome sequences in many organisms. Human genome sequencing has fallen in price from $3B in 1990 to $600 today and is expected to decrease to $100 in the near future, with some predicting that the $100 genome is not far behind [1].

Whole-genome sequencing of an individual organism represents a new reality of genomics. According to the NCBI genome database, 23,118 eukaryotic genomes have been sequenced as of 2 June 2022, including 1037 human genomes and 10,962 fungal genomes. However, many of the draft genomes contain thousands of individual sequences with no information on how these pieces are assembled into chromosomes. Long-read sequencing technologies, together with sequence scaffolding methods, have enabled the synthesis of chromosome-level de novo genome sequence assemblies. Notably, the quality of the assembled chromosome-level sequences varies among plant species and remains a challenge for large genome species [2]. The first of the largest plant genomes sequenced and assembled at the chromosome level are wheat ([3], genome size 15.8 Gb) and garlic ([4], genome size 15.9 Gb, cvalues.science.kew.org, accessed on 9 August 2022). The authors of the sequencing projects applied several sequencing technologies, including SMRT sequencing (PacBio Sequel), Oxford Nanopore Technology sequencing Illumina HiSeq paired-end, 10× Genomics, and high-throughput chromosome conformation capture (Hi-C) sequencing to construct chromosome-level reference genomes. While modern tools can produce very long reads, chromosome-level assembly is still a major bottleneck in genome sequencing projects. Advances in assembly algorithms have dramatically improved the ability to reconstruct the genomes of complex repeat-rich plants [5]. Even with sophisticated assembly algorithms, short sequencing reads make it difficult to discriminate between repeats and duplications [6,7] or ancestral polyploidy [8,9].

Genetic maps are commonly used to create chromosome-scale pseudomolecules in plants’ genome assembly, as they order and orient the assembled contigs/scaffolds [10,11,12,13]. However, even highly dense linkage maps do not cover the region of suppressed recombination, e.g., pericentromeric, knobs and subtelomeric heterochromatin [14,15,16]. Genetic maps do not carry information about the physical position of markers and the distance between them, and only give an idea of the order of markers in a linkage group. Moreover, it is impossible to assign contigs lacking DNA markers due to the inability to find polymorphism in the parental lines of a given mapping population. There are several molecular techniques that can generate physical genome maps, such as optical mapping [17,18], long-read sequencing [19], chromosomal conformation capture (Hi-C) [4,20] and the direct visualization of molecular markers on physical chromosome [21].

Onion (*Allium cepa* L.) is ubiquitously used as a flavorful and healthy food, not to mention its remarkable medicinal properties, both preventive and curative. Onion also possesses anti-cancer properties due to the accumulation of a number of bioactive compounds, including organosulfur compounds (review [22]), as well as antibiotic, anti-inflammatory, antiplatelet, antidiabetic, and cardio protective properties (review [23]). In terms of global production value, onions are ranked second after tomatoes (www.fao.org/faostat, accessed on 30 March 2022). Onion is a diploid (2n = *2x* = 16) species with a genome size of approximately 16,400 Mb/1C [24]. The first de novo assembly of the genome of a doubled haploid *A. cepa* accession was reported by Finkers and colleagues in 2021 [25]. They combined short- and long-read sequence technology (Illumina and PacBio), and the dovetail scaffolded genome assembly was anchored into pseudomolecules using four previously published genetic linkage maps [26,27,28,29]. The authors managed to assemble ∼91% of the onion genome: 14.9 Gb in 92.9 K scaffolds with a scaffold N50 size of 454 Kb. A total of 2 Gb out of 14.9 Gb was ordered into eight pseudomolecules. The large size of the genome, containing a huge number of repetitive sequences, prevented the gapless chromosome-level genome assembly and scaffold sequence ordering.

In this study, we used the ultrasensitive Tyr-FISH method to map transcriptome markers from linkage maps [29] on physical chromosomes of *Allium cepa*. In addition, the *mlh1* gene, which was not mapped on the transcriptome map, was located on chromosome 2. In total, the physical positions of 24 unique genes were established. Integrated physical, genetic and pseudochromosome genome assembly maps of chromosome 2 and 6 were constructed.

## 2. Results

### 2.1. The Development of Genome-Wide Specific Tyr-FISH Probes

In order to obtain high specific genomic Tyr-FISH probe, we developed a pipeline based on the selection of a unique sequence with minimal potential fluorescent background arising from non-specific in situ hybridization. The pipeline comprised several steps to avoid the inclusion of repetitive sequences into the probes: (1) protein non-coding transcript sequences were discarded from the analysis, as they often possess repeats; (2) the genomic sequences were further analyzed by the similarity search to the Viridiplantae repeat database (Dfam 3.3) and (3) the similarity search against the onion genome assembly was used to filter out the probe sequences with a substantial amount of genome-specific repeats.

Due to the availability of databases of whole genomic and transcript sequences of *A. cepa* [25,29], we carried out a careful in silico selection of sequences to develop genome-wide specific Tyr-FISH probes. Creating a high specific probe is especially critical for fluorescent signal detection with Tyr-FISH. The method combines the advantage of the enzymatic procedure that provides signal amplification due to the deposition of many substrate molecules, and that of fluorescence-based detection, which is higher than the absorbency used in enzymatic detection. The signal amplification up to 1000 times may result in many non-specific signals, even if the probe DNA contains a tiny sequence that is not unique.

The stringency of hybridization and washing together define the percentages of hybridized nucleotides to keep the double helix hybrid stable [30]. Based on our previous work [21] and this study, a stringency of 80–82% was optimal for a unique target visualization with a genome-wide specific probe.

The nick-translation protocol for the incorporation of labeled nucleotides into DNA was used for probe labeling. Nick translation is convenient and effective for large cloned inserts [31]. The nick-translation enzyme mixture of DNA polymerase I and DNase I produces a high incorporation of label into probe DNA, giving that the probe length is about 200–300 bp. Therefore, in situ hybridization process involves not a whole length probe sequence at once, but a set of fragments. Short probe fragments can stably hybridize to chromosomal DNA at non-specific sites, which can make it difficult to distinguish between the target signal.

Considering all of the above, the goal of the bioinformatic steps of probe design was to select the probe with the highest identity and query cover for the target site and the least amount of "dangerous" hits outside the target site. We selected the most specific probes using a simple local alignment. The most specific probe has none or the least amount of "dangerous" hits (see definition in Methods) in local alignment on genome assembly. In addition, a high level (80–82%) of stringency washing was optimally applied for a unique target visualization with a genome-wide specific probe [21].

The most marker sequences themselves are not long enough for probe design. The typical target site length for Tyr-FISH is at least 1 Kb, while the markers’ length is usually below this threshold. Therefore, we designed primers on the genomic sequence of the gene that contains the corresponding transcript-based marker. The length of gene sequences provides more space for probe design and enables to create a specific probe for a short unique target.

Among the 52 candidate genes for Tyr-FISH mapping of chromosome 2 and 86 for chromosome 6, markers corresponding to them were randomly selected from different regions of linkage groups. Based on selected markers from genetic maps, we cloned and sequenced 12 probes for chromosome 2 and 11 probes for chromosome 6. The target sequence length ranged 1018–3866 bp for chromosome 6 and 1008–2801 for chromosome 2 (Table 1). Observed sequence lengths were close or equal to expected lengths calculated from *Allium cepa* v1.2 genome assembly [25]. Probe sequences demonstrated a very high similarity (≥99%) to genomic sequences from *Allium cepa* v1.2 genome assembly [25].

In order to test the ability of Tyr-FISH for mapping any target gene that is not in a genetic map and use it as an anchor in genome assembly, we developed a probe on *mlh1* gene involved in DNA mismatch repair. As shown by Zhang et al. [32], *mlh1* is maintained as a single copy in most angiosperm species. The sequence of MLH1 protein from *Solanum lycopersicum* (GenBank: ABO07413.1) was used for the identification of the genomic sequence of *mlh1* gene in the *A. cepa* transcriptome. This transcript was then used to search for the corresponding gene in the *A. cepa* genome assembly. We found that the *mlh1* gene was located in the scaffold assigned to chromosome 2 in the *A. cepa* v1.2 genome assembly.

### 2.2. A Dual-Color Tyr-FISH Visualization of Marker/Gene

We performed the dual-color sequential Tyr-FISH with 23 selected markers and the *mlh1* gene. In each pair of probes, one probe was labeled with biotin-16-dUTP and detected with tyramide-Cy3 (red fluorescence), and the other was labeled with digoxigenin-11-dUTP and detected with tyramide-FITC (green fluorescence).

#### 2.2.1. Chromosome 2

The dual-color Tyr-FISH with genomic amplicons Unigene28076 (1466 bp) and Unigene23526 (1772 bp) revealed twin signals arising from two sister chromatids on the distal region of the short arm of both homologous chromosomes 2 (Figure 1). The relative position of hybridization sites from the centromere on the chromosome arm (RPHC) of Unigene28076 (green signals) was 60.0±2.6, and signals were found in 47.3% of the analyzed metaphases (Table 2). The Unigene23526 (red signals) was more distant from the centromere at an RPHC of 79.6±2.1 (detection frequency 57.2%) than the Unigene28076.

The dual-color Tyr-FISH mapping of the CL5148.Contig1 (1294 bp) and Unigene572 (2801 bp) genomic amplicons revealed twin signals from both probes on the short arm of chromosome 2 (Figure 2 and Appendix A). CL5148.Contig1 (green signals) was located at an RPHC 67.3±1.1 (detection frequency 45.7%). The position of Unigene572 (red signals) was at a RPHC 60.6±2.55 (detection frequency 77.4%), which is more proximal than CL5148.Contig1 (Table 2).

The dual-color Tyr-FISH with Unigene10061 (1650 bp) and CL4449.Contig1 (1231 bp) revealed twin signals from both probes on the short arm of chromosome 2 (Figure 2 and Appendix A). The Unigene10061 (green signals) was placed at an RPHC 67.7±2.4 (detection frequency 49.3%). The CL4449.Contig1 (red signals) was placed at an RPHC of 54.0±1.7 (detection frequency 42.8%), which is more proximal than Unigene10061 (Table 2).

The dual-color Tyr-FISH with Unigene25645 (1118 bp) and Unigene27326 (1008 bp) revealed twin signals from both probes on the short arm of chromosome 2 (Figure 2 and Appendix A). The Unigene25645 (green signals) was placed at an RPHC of 43.6±2.3 (detection frequency 52.6%). Unigene27326 (red signals) was placed at an RPHC of 23.9±1.6 (detection frequency 45.1%), which is more proximal than Unigene25645 (Table 2).

The dual-color Tyr-FISH mapping of genomic amplicons Unigene28713 (1323 bp) and Unigene5305 (1627 bp) revealed twin signals from both probes on the long arm of chromosome 2 (Figure 2 and Appendix A). The Unigene28713 (green signals) was located at an RPHC of 19.0±4.6 (detection frequency 52.4%). Unigene5305 (red signals) was located at an RPHC of 52.3±1.5 (detection frequency 51.6%) that was more distal than Unigene28713 (Table 2).

The dual-color Tyr-FISH with genomic amplicons Unigene23418 (1368 bp) and Unigene10683 (1160 bp) revealed twin signals from both probes on the long arm of chromosome 2 (Figure 2 and Appendix A). The Unigene23418 (red signals) was located at an RPHC of 82.9±1.6 (detection frequency 46.3%). Unigene10683 (green signals) was located at a RPHC of 69.6±5.7 (detection frequency 47.8%) that is more proximal than Unigene23418 (Table 2).

The *mlh1* gene was previously described by Zhang et al. [32] as a single copy in most angiosperm species and was located in the pseudochromosome 2 scaffold of *A. cepa* v1.2 genome assembly [25]. Tyr-FISH proved the position of the *mlh1* gene on chromosome 2. Tyr-FISH with genomic amplicon of the *mlh1* gene (1349 bp) revealed twin signals on the long arm of chromosome 2 at a RPHC of 55.5±2.1 (detection frequency 48.1%), which was very close to the previously mapped Unigene5305. To test the relative position of the *mlh1* gene and Unigene5305, the dual-color Tyr-FISH was carried out. The *mlh1* gene (red signals) was located more distally compared to Unigene5305 (green signals; Figure 2, Appendix A and Table 2).

#### 2.2.2. Chromosome 6

The dual-color Tyr-FISH with genomic amplicons of CL6133.Contig1 (1018 bp) and Unigene13813 (3809 bp) revealed twin signals from both probes on the short arm of chromosome 6 (Figure 3). The CL6133.Contig1 (green signals) was placed at an RPHC of 54.3±0.5 (detection frequency 42.1%). The Unigene13813 (red signals) was located at an RPHC of 60.7±2.9 (detection frequency 77.8%), that is more distal than CL6133.Contig1 (Table 3). It is noteworthy that the expected position of markers on the physical chromosome according to their position on the genetic map should be the distal region of the long arm (CL6133.Contig1—106.7 cM; Unigene13813—101.5 cM). According to cytogenetic nomenclature, chromosome idiograms always have a short arm at the top [33]. The graphical presentation of the genetic linkage map also follows this rule. Therefore, we may conclude that the graphical presentation of the genetic linkage group assigned to chromosome 6 [29] was upended and shown with a long arm at the top.

The dual-color Tyr-FISH mapping of genomic amplicons Unigene7941 (2286 bp) and Unigene13863 (1084 bp) revealed twin signals on the short and long arms of chromosome 6. Unigene7941 (green signals) was placed on the short arm at an RPHC of 39.4±2.3 (detection frequency 73.4%), while Unigene13863 (red signals) was placed on the long arm at a RPHC of 24.4±1.8 (detection frequency 43.6%; Figure 4, Appendix A and Table 3).

The dual-color Tyr-FISH with genomic amplicons of CL4877.Contig2 (1180 bp) and Unigene49 (1110 bp) revealed twin signals from both probes on the long arms of chromosome 6 (Figure 4 and Appendix A). CL4877.Contig2 (green signals) was placed at an RPHC of 33.7±3.1 (detection frequency 46.8%). Unigene49 (red signals) was located at an RPHC of 29.9±2.5 (detection frequency 43.6%), that is more proximal than CL4877.Contig2 (Table 3).

The dual-color Tyr-FISH mapping with genomic amplicons of Unigene10558 (1143 bp) and Unigene22659 (1457 bp) revealed twin signals from both probes on the long arm of chromosome 6 (Figure 4 and Appendix A). Unigene10558 (green signals) was located at an RPHC of 62.4±2.9 (detection frequency 49.5%). Unigene22659 (red signals) was located at an RPHC of 60.0±1.9 (detection frequency 54.8%), which is a bit more proximal than Unigene10558 (Table 3).

The dual-color Tyr-FISH mapping with genomic amplicons of CL39.Contig3 (1044 bp) and Unigene28149 (1236 bp) revealed twin signals from both probes on the long arm of chromosome 6 (Figure 4 and Appendix A). CL39.Contig3 (green signals) was placed at an RPHC of 20.0±3.0 (detection frequency 42.1%). Unigene28149 (red signals) was located at an RPHC of 72.3±2.7 (detection frequency 46.7%), which is more distal than CL39.Contig3 (Table 3).

The latest cloned amplicon from the genetic map assigned to chromosome 2 was Unigene8201 (3866 bp). The position of Unigene8201 on the genetic map was 10.9 cM. The closest marker to Unigene8201 on the genetic map was already Tyr-FISH-mapped Unigene28149 with a genetic position of 13.5 cM and a chromosomal position at an RPHC of 72.3±2.7. The dual-color Tyr-FISH with Unigene8201 (red signals) and Unigene28149 (green signals) revealed twin signals arising from both probes on the long arm of chromosome 6. The position Unigene8201 was at an RPHC of 78.7±1.6 (detection rate 86.7%), which was more distal compared to Unigen28149 (Figure 4, Appendix A and Table 3).

### 2.3. The Integration of Genetic, Cytogenetic and Pseudochromosome Maps

Chromosomal positions of Tyr-FISH mapped markers were aligned with their position on the genetic map [29] and on the corresponding pseudochromosomes in the *A. cepa* genome assembly [25]. For an appropriate comparison of genetic, chromosomal, and pseudochromosomal maps, the positions of markers on the physical chromosome were expressed as a percentage of the fractional length (FL—the distance from the telomere end of the short arm to the fluorescent signal, divided by the length of the entire chromosome).

#### 2.3.1. Chromosome 2

Among 13 markers mapped by Tyr-FISH, 8 markers were located on the short arm, and 5 markers, including the *mlh1* gene, were located on the long arm of chromosome 2. The order of markers in the genetic map of chromosome 2 and the corresponding pseudochromosome map completely coincides (Figure 5). In the de novo assembled genome of *A. cepa* [25], anchoring scaffolds into pseudomolecules was performed using multiple EST-based genetic maps, including the genetic map developed by Fujito et al. [29]. However, the visualization of markers on the physical chromosome using Tyr-FISH revealed a discrepancy between the genetic and chromosome maps and, accordingly, with the pseudochromosome assembled from the genetic map. The Unigene23526 was found first on the top at FL = 6.5% of the physical chromosome map, while it was placed second both genetic (9.0 cM) and pseudochromosome (17 Mbp) maps after Unigene28076. The Unigene10061 was located second at FL = 10.9% after Unigene23526 on the physical chromosome, while on the pseudochromosome (41 Mbp) and genetic (14.7 cM) maps, it was located fifth after Unigene572. The Unigene28076 was located first at the top of both genetic (5.9 cM) and pseudochromosome (6 Mbp) maps, while on the physical chromosome, this marker was found fifth at FL = 13.4% following the Unigene23526, Unigene10061, CL5148.Contig1 and Unigene572 markers. Thus, for three markers out of five localized in the distal region of the short arm (FL 0.0–13.4%) from the region of 0.0–17.7 cM on the genetic map, a discrepancy was revealed. Previous studies on the composition of the *A. cepa* chromosomes in this region have revealed high amounts of varies repetitive elements, including Ty1/Copia retrotransposons [34], a En/Spm-transposable element-like sequence [35], a 375 bp satellite DNA [36,37], a 314-bp tandemly repeated DNA sequence [38]. It is known that crossovers are largely suppressed in repeat-rich heterochromatic regions in plant genomes [39,40]. Study of the heterochromatic region location on the *A. cepa* chromosomes using C-banding showed that all chromosomes possess a small distally located band up to about 2–3 microns in length in colchicine metaphase chromosomes [41]. Moreover, the detection of methylated chromosome regions with antibodies to 5-mC showed that subtelomeric region bearing satellite DNA are always methylated, and next to it, the region of highly repeated DNA are often methylated in the short arm of chromosome 2 of *A. cepa* [42]. The hypermethylated region is associated with the absence of recombination [43] and is sufficient to silence crossover hot spots [44]. Thus, the discrepancy between the order of the location of markers on the genetic map and the physical chromosome is most likely associated with the suppression of recombination in this region of the chromosome. Such discrepancies between genetic and chromosome maps occurring near the distal regions of the chromosome arms because of recombination suppression were detected during the assembly of the tomato genome [15].

So, the integration of the cytogenetic map (Tyr-FISH mapping) with genetic and pseudochromosome maps revealed that among the 12 analyzed markers, 3 (25.0%) markers were placed in the wrong position on the genetic map assigned to chromosome 2, and among 13 markers, including the *mlh1* gene, 3 (23.1%) markers on pseudochromosome 2 maps.

#### 2.3.2. Chromosome 6

Among 11 selected markers, 3 markers were placed on the short arm and 8 markers were placed on the long arm. Tyr-FISH mapping of markers on chromosome 6 revealed that the genetic map of chromosome 6 was upended. However, the order of the markers in pseudochromosome 6 corresponded to the order of rotated by $180^{\circ }$ genetic map. For the alignment of genetic, chromosomal, and pseudochromosomal maps, the genetic map was turned upside down. Tyr-FISH visualization of markers on chromosome 6 showed a discrepancy between the genetic and pseudochromosome maps and the chromosome map. CL6133.Contig1 was located after Unigene13813 on the short arm of chromosome 6, while it was first at the top of both the genetic and pseudochromosomal maps (Figure 6). According to the Tyr-FISH results, the centromere is located between markers: Unigene7941 on the short arm and CL39.Contig3 on the long arm. However, on the genetic map, CL39.Contig 3 was located far from the centromere following Unigene13863, CL4877.Contig2, Unigene49, Unigene10558, and Unigene22659. Such a notable discrepancy in the pericentromere heterochromatin can be explained by the previously described suppression of crossover recombination in that region of *A. cepa* [45]. The order of CL4877.Contig2 and Unigene49 was inverted on both the genetic and pseudochromosome maps relative to their order on the physical chromosome map. The order of Unigene10558 and Unigene22659 also was inverted on the genetic map relative to their order on the physical chromosome map, but on the pseudochromosome map, the position of Unigene22659 coincided with that on the physical chromosome. The reason for this discrepancy in the order of markers on the genetic and pseudochromosome maps is explained by the use of multiple EST-based genetic maps for anchoring scaffolds into pseudomolecules [25].

Taken all together, validation of genetic and pseudochromosome map using Tyr-FISH visualization of markers on physical chromosome 6 revealed that among 11 analyzed markers, 4 (36.4%) and 3 (27.3%) markers were placed in the wrong position on the genetic map assigned to chromosome 6 and on the pseudochromosome 6 map, respectively.

## 3. Discussion

Plant genomes with large, complex genomes and varying levels of ploidy remain difficult to assemble. Illumina widely used NGS technology is responsible for sequencing a huge amount of data, which generates short sequence readouts redundantly sampled from long target molecules. PacBio Sequel and Oxford Nanopore offer vast improvements over Illumina sequencing for long reads. The increase in sequencing capacity is frequently displayed, outpacing Moore’s law [46]. Now there is no sequencing problem, but the problem of de novo genome assembly remains. Many plant sequencing projects are focused on species without an existing reference genome. Whole-genome sequencing is much more informative when linked and oriented to chromosomes than unlinked and disordered scaffolds. However, current bioinformatic tools, assembly aid methods, and long-read technologies cannot provide the routine assembly of gapless telomere-to-telomere genome sequences yet.

In this work, we showed that the visualization of markers/genes on physical chromosomes opens up a great opportunity for genome assembly improvements. *Allium cepa* v1.2 genome assembly (www.oniongenome.wur.nl, accessed on 25 March 2022, [25]) contain 86,073 gene sequence in high confidence (HC) annotation set. All of these genes show similarity to the TrEMBL database of protein sequences. 66,882 of these genes are located in 23,661 unplaced scaffolds with a total length of 8.5 Gb which is 3.5 times larger than the assembly part included in chromosomal scaffolds. According to high confidence (HC) annotation set, minimal length of gene-containing contig is 1207 bp, which can be detected by Tyr-FISH with proper frequency. However, the success of the Tyr-FISH probe design depends on the quality of the genome assembly and annotation. Genome assembly quality is represented by both completeness and contiguity (NGx and NAx characteristics), while annotation quality depends on genome assembly quality and gene prediction accuracy and represented by assembly gene space completeness. Automatic ab initio gene prediction algorithms often make a substantial errors and can jeopardize subsequent analysis, including functional annotation, identification of genes, etc. This is especially true for large genomes, where incomplete genome assembly is one of the problems [47]. The main difficulty of eukaryotic gene prediction is the presence of introns in gene sequences. This is one of the reason why the best automated gene finders are far less accurate on eukaryotes [48]. The increase in the number of exons and the deviation of the intron length from the intermediate one (50–200 nucleotides) affects the accuracy of all gene prediction programs. Long proteins, which are originated from long genes, are more likely to be badly predicted [47]. In our work, we faced two problems associated with the quality of the *A. cepa* genome assembly. First, during identification of the genomic sequences based on transcriptomic sequences, many transcripts mapped ambiguously both on chromosomes and unplaced scaffolds. Since it is not possible to understand which chromosome scaffold belongs to, genes and markers located in such scaffolds were excluded from analysis. Second, the best probe candidates for the selected genes also aligned ambiguously on several unplaced scaffolds along with specific hits on chromosome scaffold. This makes it impossible to design specific probes for a number of genes. Nonetheless, even using incomplete genome assembly, it was possible to design Tyr-FISH probes for single-copy genes and visualize them on a physical chromosome. Tyr-FISH mapping showed up to 36.4% discrepancy between genetic and cytogenetic maps. Additionally, the position of the *mlh1* gene, which was placed in pseudochromosome 2 scaffold of *A. cepa* genome assembly and was not in the genetic map, was established on physical chromosome 2. This result suggests the possibility of Tyr-FISH mapping of any gene that may not be in the genetic map or in the assembly.

ALLMAPS is one of the major steps in producing high quality genome assemblies that uses a variety of mapping information (genetic maps, Hi-C and optical mapping) to reconstruct the most likely chromosomal assemblies [49]. The program uses multiple sources of information for scaffolds anchoring and ordering that gives the opportunity to cover various parts of the chromosomal landscape in order to get better genome assemblies. Physlr pipeline provides an alternative way to build chromosome-scale physical maps using a long-range information provided by linked reads [50]. However, not all current pipelines routinely achieved high contiguity, and especially in *de novo* sequencing studies, the resulting assemblies were often highly fragmented [50,51]. In our study, to verify the assembly of the onion genome, the Tyr-FISH method was used, which showed that about 23.1% (chromosome 2) and 27.3% (chromosome 6) of the tested genes differed between physical chromosomes and pseudochromosomes. It may suggest that up to a quarter of scaffolds in onion genome assemblies may be arranged incorrectly. A previous FISH-based evaluation of tomato genome assembly revealed a similar (30%) error ratio in scaffold order [52].

In conclusion, we demonstrate that Tyr-FISH mapping of unique sequences provides useful information for evaluating and verifying plant genome assembly at the chromosome level. In situ mapping data can be utilized by ALLMAPS for producing high quality chromosome scaffold genome assemblies along with other mapping data as Hi-C and optical mapping. While Hi-C and optical mapping operate at the genome level, the Tyr-FISH method navigates a single chromosome by linking the assembled DNA sequence to cytologically identified chromosomes, chromosome arms, telomeres, centromeres, and other chromosomal landmarks. Tyr-FISH can play the role of a bridge and fill in gaps in the arranging of contigs in the representation of entire chromosomes.

## 4. Materials and Methods

### 4.1. Plant Materials

*Allium cepa* L., var. “Haltsedon” (2n = *2x* = 16) were grown in pots in a greenhouse under controlled conditions: 14 h photoperiod (REFLUX lamp 400 W; light intensity: 8000 lx) at a temperature of 20–22 °C.

### 4.2. Tyr-FISH Probe Preparation

The design of genome-wide specific probes was performed as we described earlier [21], with several modifications. Markers from linkage group 2 (198 markers) and 6 (126 markers) of transcriptome genetic map of *A. cepa* reported by [29] were used. Transcript sequences of *A. cepa* containing marker sequences were identified in *A. cepa* TSA (Transcriptome Shotgun Assembly) database using blastn 2.12.0+ [53] with parameters *-num_alignments 1 -evalue 1e–5*. Extracted transcript sequences were annotated against Uniprot release 2022_02 database using blastx 2.12.0+ with parameters *-num_alignments 1 -evalue 1e–5*. Transcripts showing no similarity against Uniprot database were excluded from analysis. Genomic sequences were identified based on the transcripts sequences. *A. cepa* genome assembly v1.2 (www.oniongenome.wur.nl, accessed on 25 March 2022, [25]) was used to extract genomic sequences of transcripts. GMAP v2021.08.25 [54] was used to extract sequences from the best mapping position (i.e., the best alignment of the transcript sequence against genome assembly) of transcript sequences from genomic assembly. Only transcripts that had the best mapping positions in chromosome 2 or 6 scaffolds according to their positions in the genetic map were included in further analysis. Genomic sequences were extracted from genome assembly by coordinates of alignment using bedtools v2.30.0 [55]. Sequences containing hardly masked fragments were discarded. A hardly masked fragments can most often be repetitive DNA sequences. RepeatMasker version 4.1.2-p1 (developed by A.F.A. Smit, R. Hubley, and P. Green; see www.repeatmasker.org, accessed on 4 May 2022) using Viridiplantae database (Dfam 3.3 as of 2020-11-09) was used for identification and soft masking repetitive sequences in genomic sequences. Primer3 v2.6.1 [56] was used for the design of primers to obtain specific probes for genomic sequences of markers. The best five pairs of primers for PCR-products 1–4 Kb long were selected for each genomic sequence. Probe sequences restricted by primer sequences were extracted from genomic sequences using seqkit v2.2.0 [57]. Probe sequences containing masked regions were discarded. Genome-wide chromosome specificity of selected probes was validated using blastn 2.12.0+ with parameters *-evalue 1e–5* and hits with identity >80% and length >100 bp were classified as “dangerous”—the ones which can lead to non-specific hybridization because of the length and identity based on stringency of hybridization and washing selected for a dual-color Tyr-FISH [21]. For each genomic sequence, the probe with none or the least amount of "dangerous" hits was selected. Information about the designed primer sequence, length of expected probe size, and annotation of transcripts corresponding to the markers in the genetic map is presented in the Appendix A.

Transcriptomic sequence of the *mlh1* gene was identified in *A. cepa* TSA database using tblastn 2.12.0+ based on protein sequence of MLH1 protein from *Solanum lycopersicum* (GenBank: ABO07413.1). The following steps for the probe preparation starting with identification of the genomic sequence in *A. cepa* genome assembly are identical to the procedure for the markers on a genetic map.

A total of 24 primer sets were used for the production of genomic amplicons. A total of 25 μL of PCR mixture contained 2.5 μL of 10× Taq Turbo buffer (Evrogen, 25 mM MgCl_2_, pH = 8.6), 0.2 mM of each dNTP, 0.2 mM of each primer, 2.5 U of Taq DNA polymerase (Evrogen, Moscow, Russia), and 100 ng of genomic DNA of *Allium cepa* L. var. Haltsedon. Amplification was performed using the following PCR program: 5 min of initial denaturation at 95 °C, 35 cycles of 95 °C 30 s, 60 °C 30 s, 72 °C 60–180 s, depending on the PCR-product’s expected length, and 10 min of final elongation at 72 °C. PCR-products were checked using electrophoresis in 1.5% agarose gel (0.5× TBE, 4 V/cm). Before cloning, PCR-product was precipitated using Evrogen Cleanup S-Cap Kit (Evrogen, Moscow, Russia) according to the manufacturer’s protocol. Concentration of precipitated PCR-product was checked using NanoDrop ND-1000 (Thermo Fisher Scientific Inc., Waltham, MA 02451 USA). TA-cloning of PCR-product was performed in a pAL2-T vector (Evrogen, Moscow, Russia), and inoculated into *E. coli* strain XL1-Blue (Evrogen, Moscow, Russia) using electroporation according to the manufacturer protocol (5:1 insert-vector ratio). The blue-white screening was used to select colonies containing a plasmid with an insert of interest. Selected colonies were screened using PCR with insert-specific primers. Colonies containing a target insert were grown overnight into 5 mL of LB medium (10 g/L tryptone, 5 g/L yeast extract, 10 g/L NaCl) and followed by plasmid DNA isolation. Plasmid DNA was isolated using an Evrogen Plasmid Miniprep Kit (Evrogen, Moscow, Russia) according to the manufacturer’s protocol. The concentration of isolated plasmid DNA was measured using the NanoDrop ND-1000 (Thermo Fisher Scientific Inc., Waltham, MA, USA). 1 μL of isolated plasmid DNA was checked using electrophoresis in 1% agarose gel (0.5× TBE, 4 V/cm). Insert in plasmid was Sanger sequenced both from 5’-end and 3’-end of the insert using standard M13 primer set (Evrogen, Moscow, Russia). The probe sequences were submitted to the NCBI GenBank system and acquired the temporary Submission ID 2596443. A total of 1 μg of plasmid DNA was labeled with digoxigenin-11-dUTP or biotin-16-dUTP using DIG or Biotin-Nick Translation Mix respectively (Roche, Mannheim, Germany) according to manufacturer protocol. Determination of the fragments length of labeled probe was checked using 2% agarose gel electrophoresis (0.5× TBE, 5 V/cm) according to the manufacturer’s protocol.

### 4.3. Chromosome Preparation

In order to arrest the chromosomes at the metaphase stage, young roots of *A. cepa* were submerged in a saturated aqueous solution of α-bromonaphthalene (1:1000, *v*/*v*) overnight at 4 °C. The root tips were fixed in freshly prepared ethanol:acetic acid mixture (3:1, *v*/*v*) for 1–2 h at RT and used for chromosome preparation according to the “SteamDrop” protocol [58] with minor modifications as follows: root tips were incubated in 0.1% (1:2:1) Cellulase Onozuka R-10 (Yakult Co., Ltd., Tokyo, Japan), Pectolyase Y-23 (Kikkoman, Tokyo, Japan) and Cytohelicase (Sigma-Aldrich Co. LLC, St. Louis, MO, USA) for 75 min at 37 °C.

### 4.4. A Dual-Color Sequential Tyr-FISH

The dual-color sequential Tyr-FISH was carried out according Kudryavtseva et al. [21]. The technique allowed to map a short unique DNA sequence on plant chromosomes and accurate determination of the physical distance between markers due to the simultaneous detection of two markers on the same chromosome with high detection frequency.

The detection of probes was performed using peroxidases (Streptavidin-HRP (PerkinElmer, Waltham, MA, USA) or anti-digoxigenin HRP (Akoya Biosciences)) and tyramides (TSA PLUS Cy3 Reagent (Akoya Biosciences, Menlo Park, CA 94025 USA) or TSA PLUS Fluorescein Reagent (Akoya Biosciences)).

Images with fluorescent signals were taken using microscope Zeiss AxioImager M2 (www.zeiss.com, accessed on 16 June 2022) with a digital Hamamatsu camera C13440-20CU (www.hamamatsu.com accessed on 16 June 2022). Image processing was performed by Zen 2.6 (blue edition) an image analysis software. The measure of Tyr-FISH signals position was performed with the program DRAWID [59]. The relative position of hybridization sites on chromosome arms (RPHC) was calculated in the form of the ratio of the distance between the site of hybridization and the centromere to the length of the chromosome arm. Karyotype analysis was performed according to the standard onion nomenclature system proposed by Kalkman [22] and confirmed by the Fouth Eucarpia Allium Symposium [60].

## Figures and Tables

**Figure 1 ijms-23-10486-f001:**
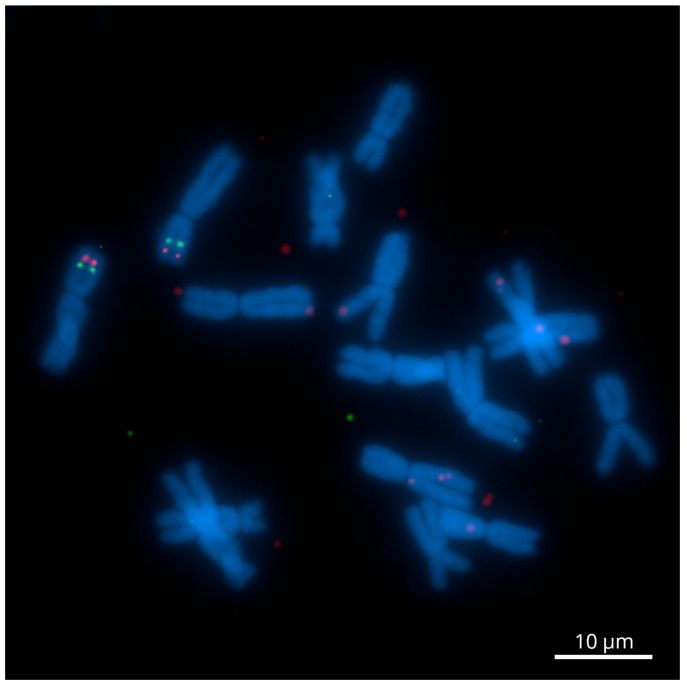
Dual-color Tyr-FISH on mitotic metaphase chromosome 2 of *Allium cepa* probing with Unigene23526 (red) and Unigene28076 (green). Scale bar—10 μm.

**Figure 2 ijms-23-10486-f002:**
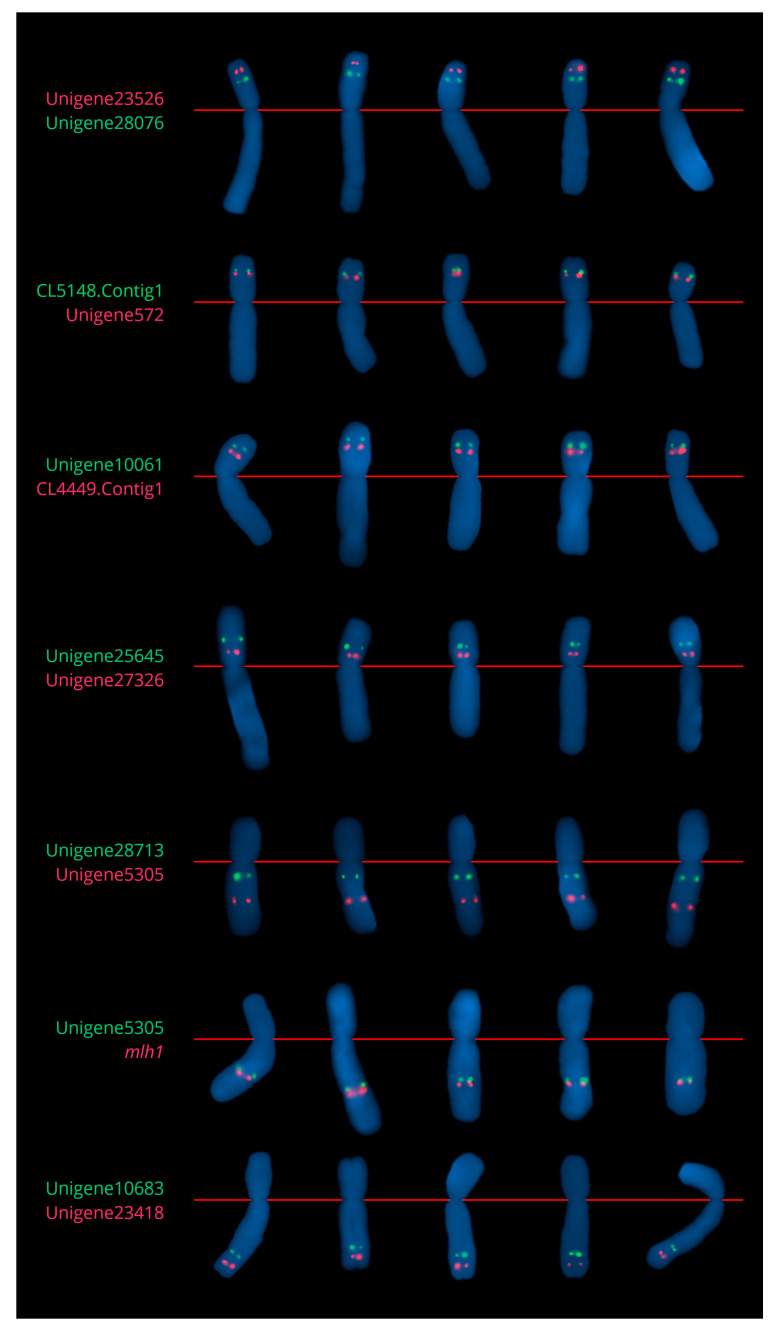
Dual-color Tyr-FISH using selected markers/gene from transcriptome map of chromosome 2. Extracted chromosome 2 from mitotic metaphase of *Allium cepa*.

**Figure 3 ijms-23-10486-f003:**
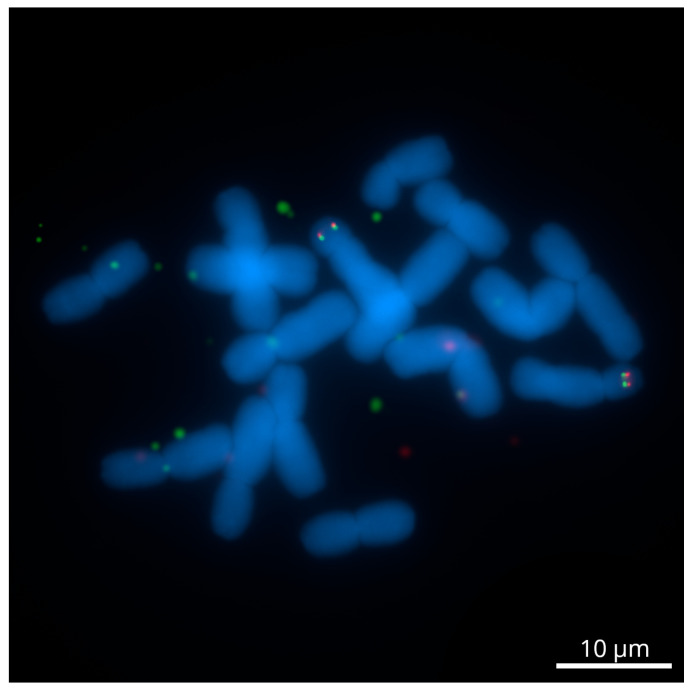
Dual-color Tyr-FISH on mitotic chromosomes 6 of *Allium cepa* probing with CL6133.Contig1 (green) and Unigene13813 (red). Scale bar—10 μm.

**Figure 4 ijms-23-10486-f004:**
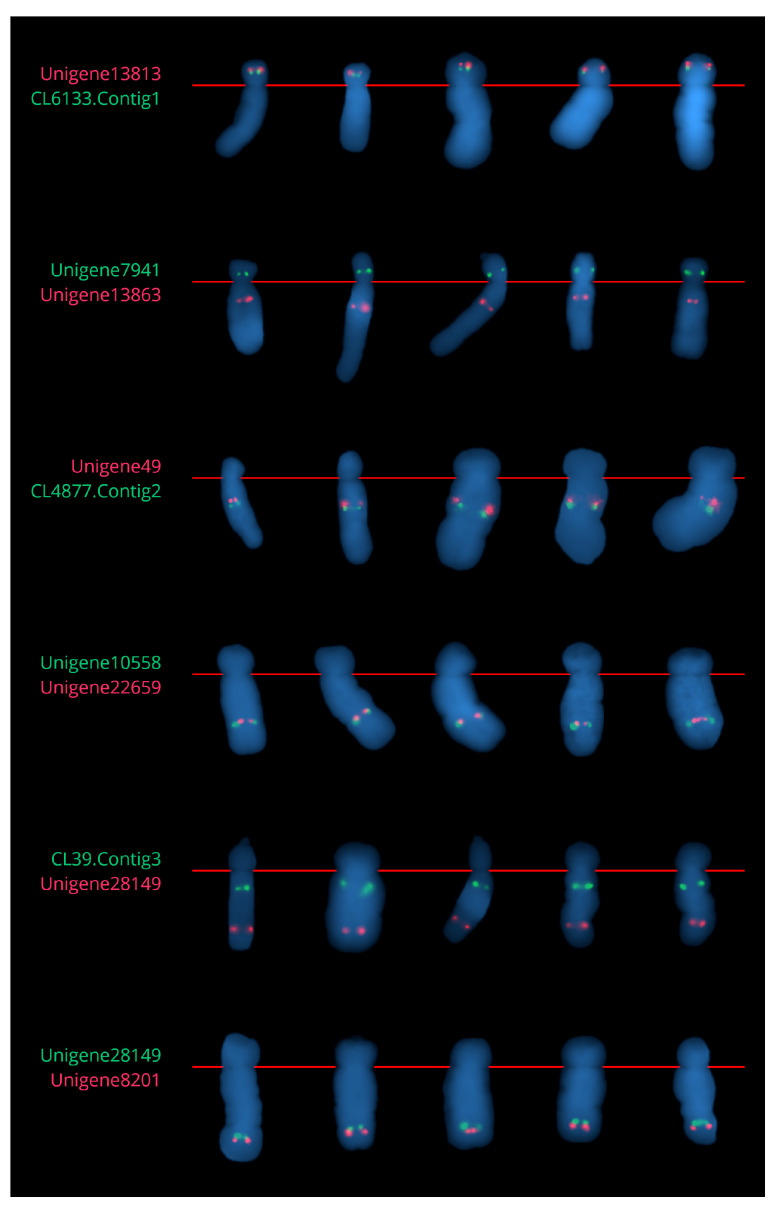
Extracted chromosome 6 of *A. cepa* from metaphases with selected markers.

**Figure 5 ijms-23-10486-f005:**
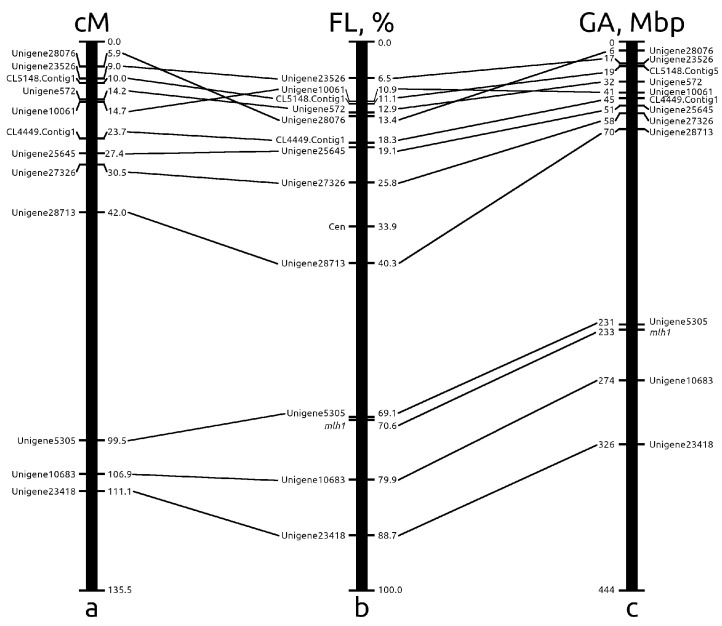
Alignment of the genetic, chromosomal and pseudochromosome maps of onion chromosome 2: (**a**) the position of Tyr-FISH markers on genetic map of the linkage group 2 described by [29]; distances in centiMorgans are shown on the right of linkage group; (**b**) the positions of Tyr-FISH mapped markers on physical chromosome are expressed as percentage of the fractional length (FL—distance from the telomere end of the short arm to the signals divided by the length of the entire chromosome); (**c**) the positions of Tyr-FISH markers on pseudochromosome 2 described by Finkers et al. [25] are expressed as the position of a gene sequence possessing of Tyr-FISH marker within one Mbp bin of pseudochromosome 2 (a total size of pseudochromosome 2–444 Mbp).

**Figure 6 ijms-23-10486-f006:**
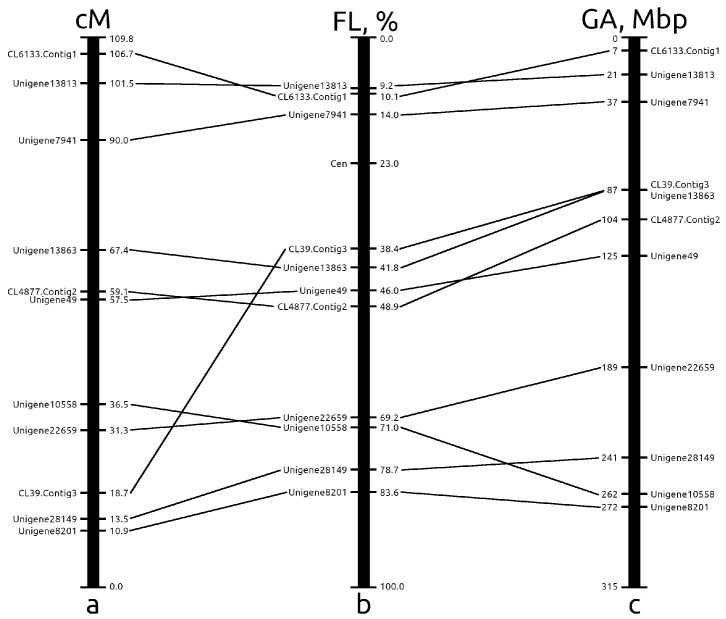
Alignment of the genetic, chromosomal and pseudochromosome maps of onion chromosome 6: (**a**) the position of Tyr-FISH markers on genetic map of the linkage group 6 described by Fujito et al. [29]; distances in centiMorgans are shown on the right of linkage group; (**b**) the positions of Tyr-FISH mapped markers on physical chromosome are expressed as percentage of the fractional length (FL - distance from the telomere end of the short arm to the signals divided by the length of the entire chromosome); (**c**) the positions of Tyr-FISH markers on pseudochromosome 6 described by Finkers et al. [25] are expressed as the position of a gene sequence possessing of Tyr-FISH marker within one Mbp bin of pseudochromosome 6 (a total size of pseudochromosome 6–315 Mbp).

**Table 1 ijms-23-10486-t001:** Comparison of the expected and observed probe lengths, including the identity of alignments between probe and genomic sequence.

Marker/Gene	Chromosome	Expected Probe Length, bp	Observed Probe Length, bp	Identity, %
Unigene23526	2	1777	1772	99.6
Unigene10061	2	1517	1650	99.7
CL5148.Contig1	2	1294	1294	100
Unigene572	2	2777	2801	99.8
Unigene28076	2	1464	1465	99.7
CL4449.Contig1	2	1231	1231	100
Unigene25645	2	1188	1188	100
Unigene27326	2	1008	1008	99.7
Unigene10683	2	1160	1160	100
*mlh1*	2	1347	1349	99.9
Unigene23418	2	1368	1368	100
Unigene28713	2	1323	1323	100
Unigene5305	2	1588	1627	99.0
Unigene7941	6	2283	2286	99.0
Unigene13863	6	1084	1084	100
CL4877.Contig2	6	1180	1180	100
Unigene49	6	1110	1110	100
Unigene10558	6	1143	1143	100
Unigene22659	6	1457	1457	100
CL39.Contig3	6	1044	1044	100
Unigene28149	6	1236	1236	100
Unigene8201	6	3866	3866	100
Unigene13813	6	3809	3809	100
CL6133.Contig1	6	1018	1018	100

**Table 2 ijms-23-10486-t002:** Relative positions and detection frequencies of Tyr-FISH signals from molecular markers on mitotic metaphase chromosome 2 of onion.

Probe	Relative Position ^1^			Detection Frequency, %	n ^3^
	Mean ± SD	Arm	n ^2^		
Unigene23526	79.6±2.1	Short	7	57.2	24
Unigene10061	67.7±2.4	Short	12	49.3	25
CL5148.Contig1	67.3±1.1	Short	9	45.7	26
Unigene572	60.6±2.6	Short	17	77.4	48
Unigene28076	60.0±2.6	Short	7	47.3	17
CL4449.Contig1	54.0±1.7	Short	6	42.8	23
Unigene25645	43.6±2.3	Short	6	52.6	26
Unigene27326	23.9±1.6	Short	14	45.1	19
Unigene28713	19.0±4.6	Long	6	52.4	16
Unigene5305	52.3±1.5	Long	14	51.6	32
*mlh1* gene	55.5±2.1	Long	8	48.1	13
Unigene10683	69.6±5.7	Long	8	47.8	23
Unigene23418	82.9±1.6	Long	5	46.3	16

^1^ the relative position of hybridization sites on chromosomes were calculated as the ratio of distance between
the site of hybridization and the centromere to the length of the chromosome arm; ^2^ number of non-overlapping analyzed chromosomes used to measure positions of hybridization signals; ^3^ number of mitotic metaphase cells analyzed.

**Table 3 ijms-23-10486-t003:** Relative positions and detection frequencies of Tyr-FISH signals from molecular markers on mitotic metaphase chromosome 6 of onion.

Probe	Relative Position ^1^			Detection Frequency, %	n ^3^
	Mean ± SD	Arm	n ^2^		
CL6133.Contig1	54.3±0.5	Short	5	42.1	17
Unigene13813	60.7±2.9	Short	9	77.8	20
Unigene7941	39.4±2.3	Short	10	73.4	14
Unigene13863	24.4±1.8	Long	7	43.6	21
CL4877.Contig2	33.7±3.1	Long	7	46.8	21
Unigene49	29.9±2.5	Long	6	43.6	22
Unigene10558	62.4±2.9	Long	6	49.5	22
Unigene22659	60.0±1.9	Long	5	54.8	21
CL39.Contig3	20.0±3.0	Long	7	42.1	18
Unigene28149	72.3±2.7	Long	6	46.7	19
Unigene8201	78.7±1.6	Long	16	86.7	15

^1^ the relative position of hybridization sites on chromosomes were calculated as the ratio of distance between the site of hybridization and the centromere to the length of the chromosome arm; ^2^ number of non-overlapping analyzed chromosomes used to measure positions of hybridization signals; ^3^ number of mitotic metaphase cells analyzed.

## Data Availability

Not applicable.

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
