# Peer review of "Integrating Genetic and Chromosome Maps of Allium cepa: From Markers Visualization to Genome Assembly Verification"

_ijms, 2022, doi:10.3390/ijms231810486_

Round 1

Reviewer 1 Report

please sequeeze the introductory lines in the abstract section and elaborate results.

rephrase line 15 

make the objectives more clear and focuse the importance of current study in the introduction section

add the conclusion section and summarize all the results

materials and methods section is choppy and needs elaboration

Reviewer 2 Report

FISH mapping is valuable for the supplement and correction of genome assembly and constructing a high-quality fine physical map. It is generally well written. Nevertheless it’s lack of sufficient introduction and discussion for FISH application in this field. I note that you have published the dual-color tyr-FISH method for plant chromosome visualization in reference 17. So what’s new in this study?
